# Investigating the Relationship between Parental Education, Asthma and Rhinitis in Children Using Path Analysis

**DOI:** 10.3390/ijerph192114551

**Published:** 2022-11-06

**Authors:** Ilaria Rocco, Giovanna Cilluffo, Giuliana Ferrante, Fabio Cibella, Alessandro Marcon, Pierpaolo Marchetti, Paolo Ricci, Nadia Minicuci, Stefania La Grutta, Barbara Corso

**Affiliations:** 1Neuroscience Institute (IN), National Research Council (CNR), 35121 Padova, Italy; 2Department of Earth and Marine Sciences, University of Palermo, 90123 Palermo, Italy; 3Department of Surgical Science, Dentistry, Gynecology and Pediatrics, Pediatric Unit, Verona University Medical School, 37134 Verona, Italy; 4Institute for Biomedical Research and Innovation (IRIB), National Research Council (CNR), 90146 Palermo, Italy; 5Unit of Epidemiology and Medical Statistics, Department of Diagnostics and Public Health, University of Verona, c/o Istituti Biologici II, 37134 Verona, Italy; 6UOC Osservatorio Epidemiologico, Agenzia di Tutela della Salute della Val Padana, 46100 Mantova, Italy; 7Institute of Traslational Pharmacology (IFT), National Research Council (CNR), 90146 Palermo, Italy

**Keywords:** asthma, rhinitis, structural equation modelling, prenatal education, children

## Abstract

Parental socioeconomic position (SEP) is a known determinant of a child’s health. We aimed to investigate whether a low parental education, as proxy of SEP, has a direct effect on physician-diagnosed asthma, current asthma and current allergic rhinitis in children, or whether associations are mediated by exposure to other personal or environmental risk factors. This study was a secondary data analysis of two cross-sectional studies conducted in Italy in 2006. Data from 2687 adolescents (10–14 years) were analyzed by a path analysis model using generalized structural equation modelling. Significant direct effects were found between parental education and family characteristics (number of children (coefficient = 0.6229, *p* < 0.001) and crowding index (1.1263, *p* < 0.001)) as well as with exposure to passive smoke: during pregnancy (maternal: 0.4697, *p* < 0.001; paternal: 0.4854, *p* < 0.001), during the first two years of children’s life (0.5897, *p* < 0.001) and currently (0.6998, *p* < 0.001). An indirect effect of parental education was found on physician-diagnosed asthma in children mediated by maternal smoking during pregnancy (0.2350, *p* < 0.05) and on current allergic rhinitis mediated by early environmental tobacco smoke (0.2002; *p* < 0.05). These results suggest the importance of promotion of ad-hoc health policies for promoting smoking cessation, especially during pregnancy.

## 1. Introduction

The prevalence of respiratory allergic diseases among children is globally high, and has been increasing throughout the last decades [1,2,3,4]. There is evidence that high parental socioeconomic position (SEP) is associated with better child health [5], although data about the relationship between SEP and respiratory allergic diseases in children is still inconsistent [6].

A low SEP has been associated with adverse housing conditions including indoor smoking [7,8], with different consequences on children’s health. Education is a good indicator of SEP [9] and is considered able to predict other proxy indicators of SEP such as income and occupation [10]. In particular, low maternal educational levels have been associated with poor general health, obesity and high risk of respiratory diseases such as asthma in children. Interestingly, mothers from low socio-economic groups were found to smoke more often during pregnancy, which was associated with asthma and weight problems in offspring [11]. Previously, the “Pollution and the Young” (PATY) study did not find a significant association between parental education and prevalence of physician-diagnosed asthma in children aged 6–12 years [11]. Highest parental education was significantly associated with higher risk of hay fever in a cross-sectional survey carried out in the municipality of Copenhagen [6]. Other studies have considered parental education as a confounder in estimating the relationship between environmental risk factors and asthma and/or rhinitis, leading to inconclusive results [12,13,14]. More recently, it has been suggested that multiple risk factors may interact and operate synergistically, leading to poor health outcomes [15]. In particular, parental education may impact on childhood respiratory health through intermediate risk factors. However, no studies have simultaneously explored the relationships among parental education, multiple risk factors and respiratory allergic diseases in children so far.

Therefore, the aim of the current study was to investigate the relationships between parental education, exposure to environmental risk factors and asthma and rhinitis in a large population-based sample of children and adolescents.

## 2. Materials and Methods

### 2.1. Study Population

The current study pooled data from questionnaires obtained in two cohorts of adolescents, living in Viadana and in Palermo, respectively.

The Viadana study is a prospective cohort study of the entire pediatric population, aged 3–14 years. Viadana is a district of Mantua, located in Northern Italy (44°55′36″ N 10°31′12″ E), about 120 km (75 mi) southeast of Milan in the Lombardia region. It has a subcontinental climate, with relatively hot summers and cool winters and a wide annual temperature range [16]. The original purpose of the epidemiological survey in the district of Viadana was investigating the respiratory health status of the pediatric population in relation to the distance of the children’s homes to the chipboard industrial facilities located in the south of study area [17]. The first wave of the survey took place in December 2006, and out of 3907 distributed questionnaires a total of 3854 were filled in and collected (response rate = 98.6%). Additional information on the Viadana study can be found in de Marco et al. [18].

The cross-sectional study conducted in Palermo involved a random sample of adolescents aged 10–17 years. Palermo is a city of 678,492 inhabitants according to the 2013 registry office, located in the northwest of the island of Sicily, on the Gulf of Palermo in the Tyrrhenian Sea (38°06′56″ N 13°21′41″ E). It has a Mediterranean climate characterized by hot and dry summers with mild temperatures for the rest of the year. The survey took place from November 2005 to May 2006. The municipality was divided into three geographical zones and 16 schools with 9922 children were identified. Of these, 2481 children were randomly selected, and complete information was available for 2150 children (response rate = 86.7%). Additional information can be found in Cibella et al. [19].

### 2.2. Exclusion Criteria

In order to identify a sample of children with homogeneous characteristics, exclusion criteria were defined before pooling the two datasets. First, children younger than 10 years and children older than 14 years were excluded (Figure 1). Then, given that the study conducted in Palermo comprised only Caucasian participants, from the Viadana sample only children whose parents were both born in Europe, West Asia or North Africa were included. Figure 1 shows the flow diagram of the sample selection process of the two studies.

### 2.3. Harmonization Process

The harmonization process of the two surveys was built upon established steps guided by the Maelstrom Research guidelines for retrospective data harmonization [20].

First step: the questionnaires and codebooks of the surveys were retrieved and analyzed by the harmonization team (BC, GC, IR, SLG). Second step: the list of variables of interest and the inclusion/exclusion criteria for the sample was created. Third step: standardized characteristics (name, label, meaning, format, categories and units) were defined for each core variable to facilitate the harmonization process across surveys datasets. The variables considered compatible were harmonized using a direct mapping when they had same question wording and categories, or through transformation or combination of variables modalities. Fourth step: two raters (IR, GC) checked independently variables formats (e.g., adequate categories created) and ranges of information (e.g., identification of outliers); in case of disagreement between the two independent raters, a third evaluation was required (SLG). Fifth step: a detailed documentation of the code used to create the harmonized variables was produced.

The steps of the harmonization process are depicted in Figure 2.

### 2.4. Outcomes

Three outcomes were created, based on internationally validated questionnaires such as those developed for the SIDRIA [12,21] and ISAAC [22] studies:

“Physician diagnosed asthma” (PDA) was defined as a positive answer to “Has your child ever had asthma diagnosed by a doctor?”.

“Current asthma” (CA) was defined as the presence of PDA plus at least one wheezing or whistling episode in the last 12 months, plus use of asthma medications in the last 12 months. A less restrictive definition for CA was also applied as the presence of PDA plus: at least one wheezing or whistling episode in the last 12 months or the use of asthma medications in the last 12 months.

“Current allergic rhinitis” (CAR) was defined as the presence, in the last 12 months, of problem with sneezing or a runny or blocked nose when the child did not have a cold or flu.

### 2.5. Parental Education

Parental education was defined as the highest educational level of the parent who filled in the questionnaire. If both parents answered the questionnaire together, the highest degree between the two was considered. When legal guardians filled in the questionnaire the record was dropped from the analysis. Parental education was then divided into two classes: “low education” (no education, primary school, lower secondary school; i.e., ≤8 years of school) and “high education” (high school, university degree, doctorate; i.e., >8 years of school), with high education as a reference category for the analysis.

### 2.6. Risk Factors

The following risk factors were considered.

The child’s body mass index (BMI) z-score was calculated according to adolescents’ height, weight, age and gender using reference data available from the WHO [23]. The +2 SD value (equivalent to BMI 30 kg/m^2^ at 19 years) was used as a cut-off for obesity.

The number of children in the family was dichotomized as “one or two” vs. “three or more”. The crowding index, dichotomized as ≤1 (less crowded) or >1 (more crowded), was defined as the number of coresidents per household, divided by the number of rooms, excluding the kitchen and bathrooms [24].

Current self-reported environmental risk factors included: level of residential exposure to traffic (high if cars and/or heavy vehicles passed constantly near home vs. middle/low otherwise), presence of mold/dampness in the child’s bedroom (yes vs. no) and of any pet in the house (yes vs. no).

Regarding exposure to smoking, four variables, categorized as present vs. absent, were considered, taking into account the time span from pregnancy to the time of the interview: exposure to maternal smoking during pregnancy; exposure to paternal smoking during pregnancy; exposure to smoke in the house during the child’s first two years of life (early environmental tobacco smoke (ETS)); and current exposure to indoor smoke due to mother/father/others smoking in the house.

### 2.7. Statistical Analysis

Quantitative data are represented as medians and interquartile range, while categorical data as absolute numbers and percentages.

To analyze the relationships between parental education, exposure to risk factors and the three outcomes of interest, a path analysis was conducted. This statistical technique is commonly used to estimate the structural relationship between multiple variables by disentangling the direct and indirect effects on the variables of interest [25,26]. Due to the dichotomous nature of the three outcomes, generalized structural equation modeling (GSEM) were applied. A logit link function was used for all the observed variables following a Bernoulli distribution and the estimated path coefficients can be interpreted analogously to regression coefficients (i.e., exponentiated coefficients are odds ratios).

Although this approach has multiple advantages, GSEM does not provide indirect effects and goodness-of-fit indicators between variables. To overcome this hindrance, the indirect effects of parental education on the health outcomes mediated by the risk factors were manually calculated. For assessment of the indirect effects, bootstrapping procedure was applied.

All models were controlled for the study (Viadana/Palermo): child’s age, gender, and parent who answered the questionnaire (mother, father or both).

Multicollinearity analysis was performed to rule out the possibility of collinearity between the variables. Variables with tolerance <0.4 (variance inflation factor >2.5) were discarded from the analyses.

Participants with at least one missing value were excluded. To investigate the potential bias from the exclusion of these participants, the main analyses were repeated, imputing missing values via multiple imputation method.

Moreover, to verify that the results were not influenced by the chosen cut-off, sensitivity analyses were carried out:Replacing the dichotomous variable on crowding index (>1) with the corresponding quantitative variable;Changing the cut-off for number of children with “one child” vs. “two or more children” and also replacing it with the corresponding quantitative variable;Changing the cut-off for parental education considering “low education” as no education and primary school, and “high education” as lower secondary school or above. Furthermore, an additional analysis was repeated considering “low education” as no education and up to high school, while “high education” as university degrees and doctorates.

All statistical analyses were performed using STATA (version 13) and SAS (version 9.4), and a *p*-value of <0.05 was considered as statistically significant.

### 2.8. Ethics Statement

The Viadana study protocol was approved by the local ethics committee of the National Health Service Mantua (Azienda Sanitaria Locale di Mantova, Italy).

The Palermo study was approved by the Institutional Ethical Committee (University Hospital of Palermo, Italy). All parents of the invited children signed a written informed consent form.

## 3. Results

### 3.1. Characteristics of Study Participants

Table 1 shows the distribution of children’s and family characteristics, environmental factors and smoke exposure of the pooled sample (*N* = 2687). The questionnaires were mainly filled in by the mothers (61.1% mother only and 30.1% both parents) and almost half of them (53.3%) had completed at least high school. The sample was balanced in gender (52.1% males), with a median age of 12 years [Q1–Q3: 11–13] and 16.3% children were categorized as obese.

Children who reported PDA were 9.2%, while those using medications to treat their condition were 5.6%. Considering those that in the last 12 months had at least a wheezing or whistling episode, this led to a 2.5% of the sample with CA, while considering a less restrictive definition for CA the percentage increased to 5.1%. A total of 23.3% of the children had CAR.

Looking at the family characteristics, most of the children had at least one sibling (82.4%) and 34.3% had at least one room at their own disposal. Exposure to environmental factors were 19.2% for high traffic, 13.0% for mold/dampness in the child’s bedroom and 24.0% for pets. Considering passive smoke exposure, 9.8% of the mothers smoked during pregnancy, while almost half of the fathers (47.0%) smoked instead. About 29.6% of the children were exposed to early ETS and 37.8% had somebody that currently smoked into the house.

### 3.2. Effects of Parental Education on Risk Factors

Figure 3, Figure 4 and Figure 5 show the path diagrams of the GSEM models concerning the relationships between parental education, risk factors and the three explored outcomes.

The educational level attained by parents did not influence either the likelihood of obesity for their children (coefficient t = 0.1207, *p* = 0.277) nor the environmental characteristics of the place where they lived (traffic: 0.0154, *p* = 0.878; mold/dampness: 0.0085, *p* = 0.942; pets: −0.0172, *p* = 0.855).

A lower parental educational level was associated with the overcrowding of the dwelling (1.1263, *p* < 0.001), as well as with a higher number of children in the household (0.6229, *p* < 0.001). Moreover, the lower educational level of parents strongly affected their smoking behaviors across their whole lifespan. Indeed, it was associated with higher exposure to passive smoke during pregnancy (maternal: 0.4697, *p* < 0.001; paternal: 0.4854, *p* < 0.001), during the two first years of children’s life (0.5897, *p* < 0.001) and currently (0.6998, *p* < 0.001).

### 3.3. Effects on PDA

Although the parental education level significantly influenced the number of children in the family (0.6229, *p* < 0.001) and the crowding index (1.1263, *p* < 0.001), the indirect effects of parental education on PDA, mediated by these factors, were not significant (respectively 0.1821, *p* = 0.164; 0.0076, *p* = 0.972) (Figure 3).

Moreover, even though the parental educational level did not directly affect the risk of their children to receive PDA (0.0384, *p* = 0.816), it had an indirect effect mediated by the maternal smoking during pregnancy (0.2350, *p* = 0.021).

**Figure 3 ijerph-19-14551-f003:**
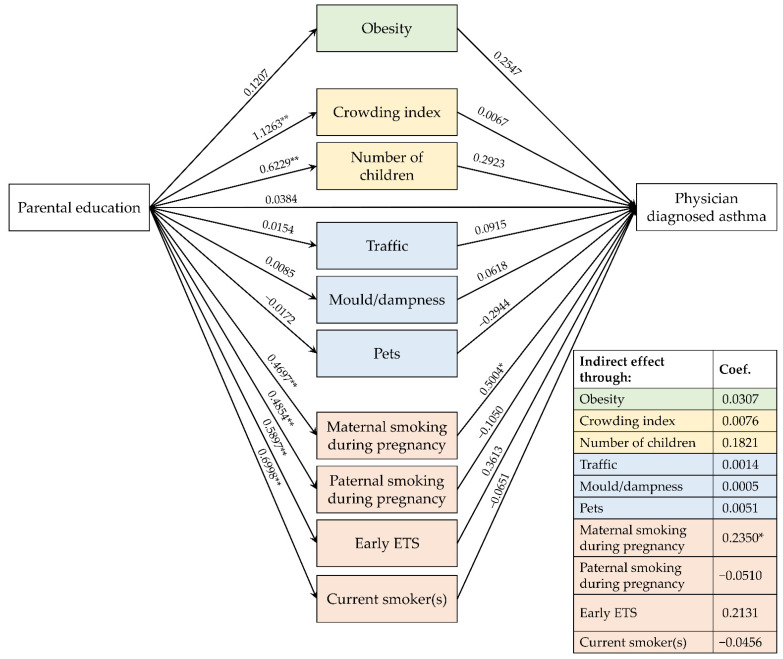
Path diagram of the relationships between parental education, risk factors and physician-diagnosed asthma. Note. ETS: Environmental tobacco smoke; * *p* < 0.05; ** *p* < 0.01.

### 3.4. Effects on CA

No significant direct nor indirect effects were found for parental educational level and CA (Figure 4). However, it is still worth noting the indirect effect of parental education on CA was mediated by maternal smoking during pregnancy even if it did not reach statistical significance (0.3384, *p* = 0.098).

Even when a less restrictive definition of CA was applied, no significant direct nor indirect effects were found for parental educational level. In this case, it needs to be highlighted that the presence of a direct effect of early ETS (0.5041, *p* = 0.051) and an indirect effect of parental education were mediated by early ETS (0.2972, *p* = 0.095), even if they did not reach statistical significance (Appendix A).

**Figure 4 ijerph-19-14551-f004:**
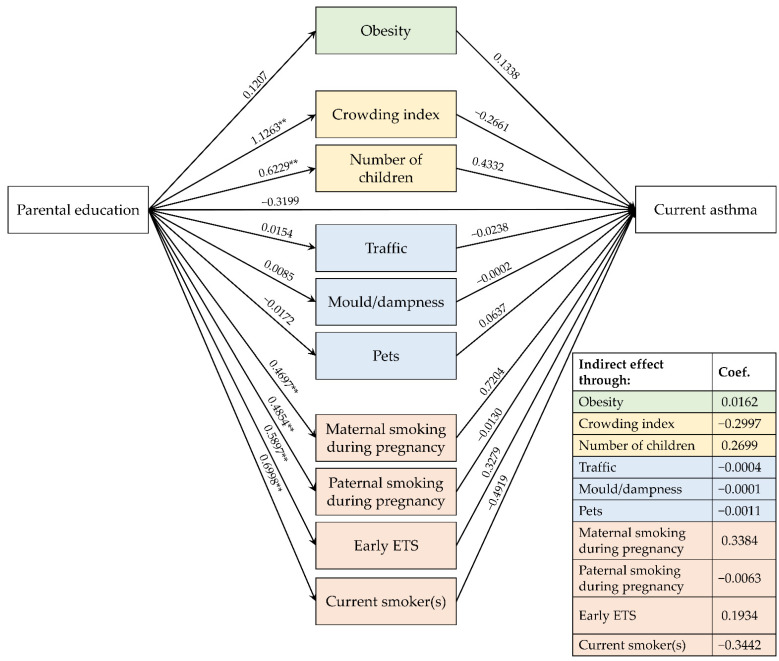
Path diagram of the relationships between parental education, risk factors and current asthma. Note. ETS: environmental tobacco smoke; ** *p* < 0.01.

### 3.5. Effects on CAR

Children’s risk of suffering from CAR was higher if mold/dampness were present in their bedroom (0.4024, *p* = 0.008) and if exposed to early ETS (0.3395; *p* = 0.020) (Figure 5).

Even though parental education level did not influence directly the risk of CAR (−0.0256; *p* = 0.823), it had an indirect effect mediated by early ETS (0.2002; *p* = 0.027).

**Figure 5 ijerph-19-14551-f005:**
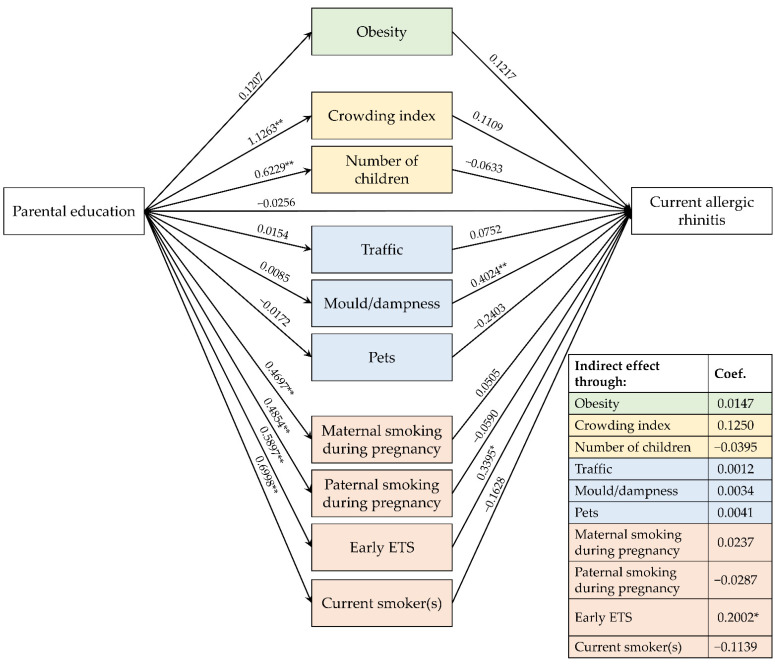
Path diagram of the relationships between parental education, risk factors and current allergic rhinitis. Note. ETS: environmental tobacco smoke; * *p* < 0.05; ** *p* < 0.01.

### 3.6. Sensitivity Analysis

The results were confirmed when analyses were repeated after multiple imputation of missing values (Appendix A), and when the discrete quantitative variable for the crowding index was used instead of the dichotomous one (Appendix A).

When the number of children in the family was dichotomized as “one child” vs. “two or more children”, the results were confirmed but the direct effect of parental education on number of children was no more significant (0.1249, *p* = 0.324) (Appendix A). Interestingly, when the discrete quantitative number of children in the family was used, a significant direct effect of it was observed on PDA (0.1947, *p* = 0.030), and the indirect effect of parental education through the number of children was borderline significant for PDA (0.0481, *p* = 0.057) and for CA (0.0663, *p* = 0.050) (Appendix A).

Changing the cut-off for parental education had some drawback. When the cut-off was set to no education plus primary school vs. lower secondary school or above, the direct effect of parental education on paternal smoking during pregnancy was no more statistically significant (−0.0434, *p* = 0.764) (Appendix A). The indirect effect of parental education through maternal smoking on PDA and through early ETS on CAR were weakened (0.2993, *p* = 0.059 and 0.1315, *p* = 0.60, respectively) (Appendix A). Modifying the categorization for parental education as no education and up to high school vs. university degree plus doctorate, the direct effect of parental education on number of children was no more significant (0.2908, *p* = 0.052), but the indirect effects of it through maternal smoking on PDA and through early ETS on CAR were preserved (0.4303, *p* = 0.049 and 0.244, *p* = 0.015, respectively) (Appendix A). No significant changes were observed for CA (Appendix A). In light of these results, it should be pointed out that the two categories of parental education with higher percentages were lower secondary school (38.2%) and high school (42.7%). For this reason, parental education was defined keeping separate those two categories; bringing them together resulted in an unbalanced variable for parental education (with only 8.5% and 10.6% in a category, respectively).

## 4. Discussion

Through path analysis, we revealed an indirect effect of parental education on PDA in children mediated by maternal smoking during pregnancy and on CAR mediated by early ETS. We also found that parental education did not directly affect the risk of their children to receive an asthma diagnosis nor to have CAR, and that although it significantly influenced the number of children in the family and the crowding index, the indirect effects mediated by these risk factors were not significant on those outcomes. No significant direct or indirect effects of parental education were found on CA.

Socioeconomically deprived groups usually face increased exposure to environmental risk factors, which may enhance susceptibility to the development of allergic respiratory diseases. Indeed, in a prospective birth cohort study an inverse relation between SEP and asthma and rhinitis has been found in children at age four [27]. It can be argued that parents with lower SEP often adopt unhealthy behavior like smoking during pregnancy and continuing to do so afterwards [28]. On the other hand, the favorable life-style of parents with high SEP appears to prevent the development of respiratory allergic disorders in their children, such as aeroallergen sensitization, lifetime prevalence of hay fever and asthma [29]. Parental education is a proxy of SEP, and it has been used as an indicator of adult SEP [9].

We found that parental education did not directly affect the risk of children to receive an asthma diagnosis. Our results are in line with those of the PATY study that did not find a significant association between parental education and prevalence of PDA in children aged 6–12 years [11]. However, we revealed an indirect effect of parental education on PDA in children mediated by maternal smoking during pregnancy. Exposure to maternal smoking in utero has been clearly associated with PDA in early life [30,31]. There is also evidence that association between maternal smoking during pregnancy and PDA in children was significant even when adjusting for parental education [32]. However, to the best our knowledge, no studies have extrapolated indirect effect of parental education, mediated be, on PDA.

Although the parental education significantly influenced the number of children in the family and the crowding index in the current study, its indirect effects on PDA, mediated by these factors, were not significant. Previous studies indicated an association of household crowding with asthma [33], likely due to a high exposure to indoor allergens in crowded households. Despite differences in study design and population, other studies failed in finding a statistically significant association between crowding and PDA [34]. These results could suggest that parental education may weaken the association between crowding and PDA, but our data do not support this hypothesis.

No significant direct or indirect effects were found for parental education and CA. This might be ascribed to the low prevalence of CA in our study population (2.5%). Indeed, this variable was defined as the presence of PDA plus at least a wheezing or whistling episode in the last 12 months, plus use of asthma medications in the last 12 months. Nonetheless, in line with our results, a previous population-based cohort study [35] did not find any association between family SEP and asthma in school-aged children after correcting for other sociodemographic factors including a wide range of family SEP indicators such as parental educational level, net household income, financial difficulties, parental employment status and ethnic background. Noteworthily, prevalence of CA in that study population was higher than in our study (5.9%), likely because CA was defined as ever PDA combined with wheezing and use of asthma medications in the last 12 months [35].

In addition, in our study parental education did influence the risk of CAR through an indirect effect mediated by early ETS. Indeed, in a previous study on school-aged children, symptoms of CAR were more prevalent in children whose mother or father had not received university-level education [36]. However, this study cannot be properly compared with the current one, due to differences in design and population; furthermore, prevalence of CAR was higher than those reported in our study (40.0% vs. 23.3%). Conversely, the prevalence of allergic rhinitis symptoms increased from lowest to highest SEP in a population-based survey on pupils aged 13–14 years [37]. Similarly, a parental bachelor’s degree was found to be a risk factor for allergic rhinitis in a web-based survey on 75,643 adolescents, likely thanks to better access to medical services with improved disease detection and raised disease diagnosis rate [38]. Again, differences in study design, population and outcome investigated make hard a comparison with the current findings.

One of the major limitations of the current study is the cross-sectional design, that did not allow to establish a temporal sequence relationship among risk factors and selected health outcomes. However, the problem of temporality does not seem be attributable to the first part of the causal chain, since parental education was attained previous than outcome assessment. The second important limitation is that the information on respiratory allergic outcomes was self-reported and was not verified by objective assessment. However, information on symptoms was provided through standardized and validated questionnaire-based definitions. At last, although several studies used education of the parents as a proxy for SEP, this variable does not necessarily reflect the SEP of children in the household. Indeed, it should be pointed out that measures of parental SEP may account for only part of the effects of social disadvantage on child health. Another possible variable that may affect the social disadvantage on child health could be parental occupational status. However, this information was available only in one of the two studies considered in the present work, so it was not possible to estimate the potential effect of parental occupation.

To our knowledge no studies have investigated the relationship between parental education and respiratory allergic diseases in childhood by means of path-analytical models. The major benefit of path analysis is that it efficiently estimates both the direct and indirect effects of the mediation process [39]. Furthermore, this approach allowed us to simultaneously consider each variable as both a predictor and an outcome according to its role in each part of the model, thus making a more realistic overview of the phenomenon of interest [25]. Another strength of the current study is the large, population-based sample. This afforded the use of a model in which many relationships among selected variables were assessed simultaneously. Indeed, pooling the data was a valuable strategy allowing us to address research questions that the individual surveys could not answer. Notably, the harmonization process of the two surveys was built upon established steps guided by the Maelstrom Research guidelines for retrospective data harmonization [20]. Lastly, multiple imputations were used to address missing information, but results did not change in the sensitivity analysis.

## 5. Conclusions

In summary, by means of path analysis, we were able to reveal an indirect effect of parental education on PDA in children mediated by maternal smoking during pregnancy. This result adds more evidence that social disadvantage on childhood asthma encompasses environmental risk factors including tobacco smoke exposure in utero.

Health policies must focus on smoking cessation especially during pregnancy, which is a vulnerable window of exposure [40]. In this context, ad-hoc training courses for promoting smoking cessation should be implemented among primary care physicians [41]. The present study identifies preventable risk factors for which there are available and effective evidence-based interventions.

## Figures and Tables

**Figure 1 ijerph-19-14551-f001:**
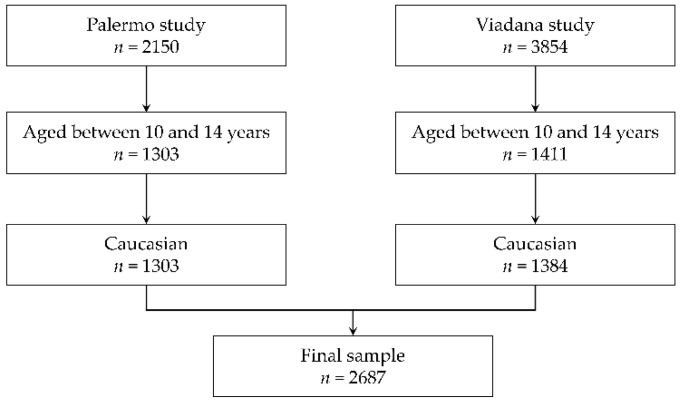
Flow diagram of the sample selection process based on exclusion criteria.

**Figure 2 ijerph-19-14551-f002:**
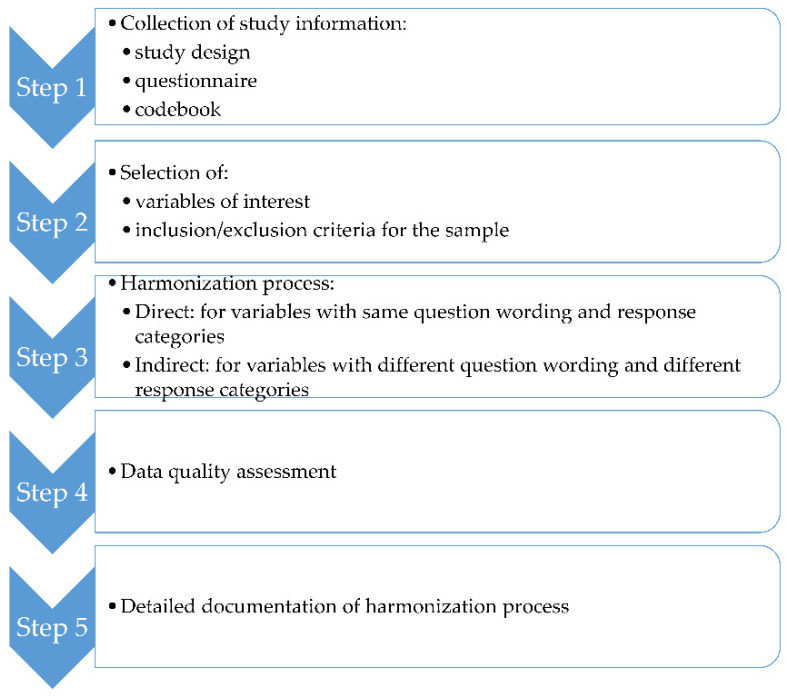
Steps of the harmonization process.

**Table 1 ijerph-19-14551-t001:** Description of the pooled sample.

	Sample
	*N* = 2687
Person who answered the questionnaire, *n* (%)	
Mother	1604 (61.1)
Father	233 (8.9)
Both parents	789 (30.1)
Parental education † (high education), *n* (%)	1365 (53.3)
**Children’s characteristics**	
Gender (male), *n* (%)	1400 (52.1)
Age (years), median (Q1–Q3)	12 (11–13)
BMI z-score, median (Q1–Q3)	0.76 (−0.13–1.63)
Obese (yes), *n* (%)	408 (16.3)
Acute respiratory diseases *(bronchitis, asthmatic bronchitis, bronchiolitis, pneumonia)* in the first 2 years of life (yes), *n* (%)	907 (34.9)
Wheezing or whistling in his/her chest at any time in the past (yes), *n* (%)	577 (21.6)
Wheezing or whistling in his/her chest in the last 12 months (yes), *n* (%)	141 (5.3)
Physician diagnosed asthma (PDA) (yes), *n* (%)	244 (9.2)
Asthma medications in the last 12 months (yes), *n* (%)	145 (5.6)
Current asthma *(more restrictive definition)* ‡ (yes), *n* (%)	66 (2.5)
Current asthma *(less restrictive definition)* § (yes), *n* (%)	137 (5.1)
Current allergic rhinitis (*presence of sneezing or a runny or blocked nose without a cold or flu in the last 12 months*) (yes), *n* (%)	620 (23.3)
**Family characteristics**	
Number of children, median (Q1–Q3)	2 (2–3)
Crowding index (≤1), *n* (%)	894 (34.3)
**Environmental factors**	
High traffic level near home (yes), *n* (%)	513 (19.2)
Mould/dampness in the child’s bedroom (yes), *n* (%)	341 (13.0)
Pets (yes), *n* (%)	612 (24.0)
**Smoke exposure**	
Maternal smoking during pregnancy (yes), *n* (%)	261 (9.8)
Paternal smoking during pregnancy (yes), *n* (%)	1225 (47.0)
Early ETS in the first two years of life (yes), *n* (%)	756 (29.6)
Current smoker(s) (somebody smoking indoor), *n* (%)	990 (37.8)

Note: † Parental education: highest educational level of the parent who filled in the questionnaire. In the case both parents answered the questionnaire together, the highest degree between the two was considered; BMI: body mass index; PDA: physician-diagnosed asthma; ‡ PDA and at least one asthma symptom or having an asthma attack in the last 12 months, and currently taking asthma medications; § PDA and at least one asthma symptom or having an asthma attack in the last 12 months, or currently taking asthma medications; ETS: environmental tobacco smoke; Q1: first quartile; Q3: third quartile.

## Data Availability

The data that support the findings of this study are available from the corresponding authors upon reasonable request.

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
