# Peer review of "Investigating the Relationship between Parental Education, Asthma and Rhinitis in Children Using Path Analysis"

_ijerph, 2022, doi:10.3390/ijerph192114551_

Round 1

Reviewer 1 Report

see attachment

Reviewer 2 Report

Review the manuscript: ijerph-1982431.

In recent years, exposure to tobacco smoke in the environment represents an important public health risk, especially in Serbia, which has a large number of smokers. The usual constituents of tobacco smoke are particles with an aerodynamic diameter of less than 10 µm (PM10), which are the most dangerous air pollutants for health. It is necessary to inform parents about the health consequences of exposure to tobacco smoke during childhood and to encourage them to change their smoking habits in order to. reduce their children's exposure to tobacco smoke at home.  

The topic of the work is generally known and has been clearly defined and proven in many researches so far.

* line 67-78 - Are there studies that are more recent than this one? It could be added if there are exist
* line 131 - WHO should be cited in references or maybe illustrate this sentences of BMI like Figure
* line 306-307 - Could you please explain even more what type of respiratory allergic disorders it could prevent?
* Table 1 - notes about signs should be described in main text or maybe make a legend in the Table
* Figure 3,4,5 - resolution of figure 3, 4 and 5 should be better with point on numbers at the lines
* https://doi.org/10.1371/journal.pone.0250255 This article presents a similar topic and you can use it as a comparative analysis
A real contribution to science would be for the authors to do a comparative analysis after education.

The analysis would show whether the education led to a reduction in exposure to tobacco smoke.

Reviewer 3 Report

Review IJERPH 2022

(International Journal of enviromental research and public health)

Investigating the relationship between parental education, asthma and rhinitis in children using path analysis, by Ilaria Rocco et al.

In this Italian study, two population-based cross-sectional studies of children and adolescents were pooled. The direct and indirect effect of parental education on asthma and rhinitis were investigated, and several personal and environmental risk factors were taken into account in the path analyses. The research question is relevant and interesting and worthy of investigation. Although I am not a statistician and not really familiar with the path analysis, the methods and result presentation seems sound and accurate. However, there are some concerns:

11.       Overall: the term ‘doctor diagnosed asthma’ is rather colloquial. Consider changing it to ‘physician-diagnosed asthma’.

22.       Overall: there are some grammatical errors and questionable wordings here and there. The manuscript would benefit from undergoing language revision.

33.       Abstract: Please specify how the outcome measure is presented (see also point 10).

44.       Introduction: Historically and in some geographical areas there still may be an inverse association between socio economic position and allergy – where those in higher SEP are more likely to have allergic rhinitis, as is also mentioned in the discussion. Therefore I suggest that the direction of the association is presented more clearly, for instance in the sentence starting with “Parental education was significantly…”

55.       Moreover in the same sentence as mentioned in point 3, a minor issue: the reference number 6 does not have brackets in the end of the sentence.

66.       Materials and methods, 2.1: Please add a short description of the two study areas. Are there any major differences in cultural, geographical, socio-economical or demographical aspects?

77.       Materials and methods, 2.2: Although not many non-Caucasian children were excluded, it is not clear why they were excluded. Did you perform any analyses where these 27 children were included? Or would it be possible to include the variable ‘parents born outside Europe, west Asia or north Africa’ in the path analyses? This aspect could also be a socioeconomic marker.

88.       Materials and methods, 2.5: many readers may not be familiar with the Italian school system, please add information on how many years of school a high school degree implies.

99.       Materials and methods, 2.6: please clarify the categorization of self-reported traffic exposure. Did the responders state whether they regarded their exposure as ‘low’, ‘medium’ or ‘high’, or were there any more specific measures? Such as a road with heavy traffic close to the home?

110.   Materials and methods, 2.7: Many readers may not be familiar with the path analysis. Please specify how the outcome measure is expressed. Is the coefficient a correlation coefficient? If so, is it possible for it to take on a value above 1 (as seen in Figure 4, for instance)? Please help us understand the outcome measure.

111.   Table 1: Only one decimal is needed to present the prevalence estimates of the demographical and environmental risk factors.

112.   Figure 3, 4 and 5: in the notes there are two footnotes * and **, but in the graphs, there is another sign after the numbers to the right, a cross. Please specify in the footnotes what the cross means.

113.   5. Conclusions: the last sentence is rather far from the content of the present study. For instance, it did not evaluate the effect of ad-hoc training courses, it did not evaluate which caregiver should give the support (more caregivers than physicians can provide smoking cessation support to pregnant women), or the level of health care (pregnant women may also be at secondary care level). Instead, highlight for instance that the present study identifies preventable risk factors for which there are available and effective evidence based interventions.
